# LEARNING GRAPH REPRESENTATION FOR MODEL ENSEMBLE

## ABSTRACT

We introduce, LGR-ME (Learning Graph Representation for Model Ensemble), a groundbreaking approach within the domain of general-purpose learning systems. Its primary focal point is to establish a foundational framework that facilitates self-adaptation and versatility in the ever-evolving landscape of emerging machine learning tasks. Despite the strides made in machine learning, it has yet to reach the adaptive and all-encompassing cognitive prowess demonstrated by biological learning systems. This discrepancy is particularly pronounced in the sphere of replicating learning representations and mastering a diverse spectrum of general-purpose learning algorithms. Our proposition entails a graph-centered representation of machine learning models. This representation operates on a graph composed of models, where the interconnections among akin models are established based on model specifications and their corresponding performances. In pursuit of this representation, we employ a graph neural network to undergo training. In this aspect, we present a novel method through the utilization of the top $k$ maximum spanning trees. This encoding is then subjected to training by a meta-model that minimizes a newly devised loss function. This combined loss function effectively accounts for both diversity and accuracy. Furthermore, we provide a theoretical examination of both the graph encoding algorithm and the newly introduced loss function. This advanced training process engenders an understanding of the intricate interdependencies and correlations existing among the model ensemble. The acquired features are subsequently harnessed to generate the ultimate output for the initial task at hand. By means of extensive empirical comparisons, we showcase the efficacy of LGR-ME in contrast to solutions predicated on ensemble pruning techniques.

## 1 INTRODUCTION

In the forthcoming era of an intricately interconnected digital landscape, the operational paradigm of computer systems will undergo a paradigm shift, necessitated by a diverse array of machine learning architectures meticulously crafted to cater to an extensive spectrum of pragmatic applications and contextual circumstances Kim et al. (2022). Regrettably, the performance of these architectures is not uniform across all datasets; rather, certain architectures excel under specific conditions while others falter. In response to this intricate challenge, ensemble methods emerge as a cohort of robust machine learning models characterized by the orchestration and amalgamation of multiple foundational learners, all oriented towards the attainment of a singular learning objective. Manifesting state-of-the-art efficacy across a myriad of applications, ensemble methods routinely outpace individual classifiers by a substantial margin in terms of performance metrics Zhong et al. (2020); Gomes et al. (2017); Yang et al. (2023). Nonetheless, the adoption of ensemble methods presents two primary challenges. The foremost challenge emanates from their resource-intensive nature, consuming significant computational time and memory resources. This is due to the necessity of loading and executing all constituent models within the ensemble during the inference phase, which can lead to considerable latency. The second challenge pertains to the potential adverse impact introduced by one or more models within the ensemble. The collaborative dynamics of the ensemble can occasionally result in certain models exerting a counterproductive influence on the collective learning process, thereby impeding the overall performance enhancement that ensemble methods are designed to achieve. Ensemble methods combine multiple individual models to make more accurate predictions

or classifications compared to using any single model alone Liu et al. (2023). However, they consider models in the ensemble as independent elements which yields to the development of intuitive fusion strategies in the research literature including voting, averaging, stacking, bagging, and boosting.

## MOTIVATIONS

> *We hypothesise that an examination of the diverse patterns present among the constituent models within the ensemble possesses the capability to yield practical insights conducive to enhancing the comprehensive learning endeavor.*

Our hypothesis suggests that by thoroughly representing and analyzing the models within an ensemble, we can identify the best subset of models that positively contribute to the learning process. This perspective aligns with the idea that not all models in an ensemble might be equally valuable or effective for a given task, and by selecting the optimal subset, we could potentially improve the overall performance of the ensemble. In this research work, we will study the dependencies among models in the ensemble via graph representation. A graph is a data structure that consists of vertices and edges that connect these vertices. Graphs are used to represent relationships or connections between different entities, and they can be powerful tools for analyzing and deriving patterns among these entities Peng et al. (2023). The idea behind it is to learn a robust representation that give valuable insights of the behaviors of the different models in the ensemble.

## CONTRIBUTIONS

This study marks the first comprehensive effort to thoroughly explore a graph representation in ensemble model. This approach is designed to effectively tackle the challenges presented by modern ensemble model. In simpler words, it introduces the novel idea of LGR-ME, serving as a foundational paradigm for creating adaptable machine learning systems. These systems aim to overcome limitations found in both dedicated single-purpose models and multipurpose models tailored for specific tasks. The main contributions of this research can be succinctly outlined as follows:

1. We propose a graph-based representation of the ensemble model to capture the different features of the models specification in the ensemble. In this context, various strategies have been developed to create the graph of models.

2. We use Graph Convolutional Neural Network as a meta model to learn how to execute a task. The GCNN minimizes a newly formulated loss function. This composite loss uniquely captures diversity, accuracy and tree structure hierarchy. We further give a theoretical analysis of the proposed loss function.

3. We conduct a comprehensive and rigorous experiments to assess the different components of LGR-ME using well-established benchmarks. The results demonstrate that LGR-ME outperforms the baseline solutions in terms of output quality, even when the components of the graph are weak in executing a task such as classification.

## 2 RELATED WORK

Ensemble learning embodies a robust learning framework that amalgamates multiple diverse learning models with the aim of enhancing the accuracy and robustness of singular-based models. This approach can be systematically categorized into five principal classes: 1) **Bagging** Zhu et al. (2022); Westfechtel et al. (2023), within this framework, a multitude of replicated instances of a solitary model undergo individual training regimens employing dissimilar subsets of the training dataset. The ultimate synthesized output is then obtained through the process of aggregating the disparate outputs generated by each constituent model. 2) **Boosting** Zhou et al. (2023); Wang et al. (2022); Gabidolla & Carreira-Perpiñán (2022), which involves the amalgamation of numerous subordinate learners to establish a proficient learning entity. The subordinate learners, characterized by their limited individual learning capabilities, are sequentially trained, with each successive model emphasizing the rectification of data instances that were inadequately addressed by its predecessors. 3) **Stacking** Dong et al. (2022); Iniyan et al. (2023); Chen et al. (2023), where a plurality of models undergoes training, and their resultant outputs are subjected to fusion via an additional model known as a meta-model.

The meta-model is specifically trained employing the outputs of the underlying models, thereby giving it the capacity to assimilate and leverage their individual strengths through an integrative learning process. 4) **Blending** Lu et al. (2019); Pu et al. (2022); Zhu et al. (2023), where predictions from multiple models are combined using a simple rule for merging the partial outputs. It involves training diverse models on the same dataset and then aggregating their decisions to make a final output. Blending is effective in reducing overfitting and improving generalization by leveraging the strengths of different models. It does not require an additional meta-model training step like stacking, making it a straightforward approach for ensemble learning. 5) **Bayesian Averaging** Chipman et al. (2006); Fersini et al. (2014); Seyedan et al. (2022) which employs a probabilistic approach to combine decisions from different models. It estimates posterior probabilities for each class based on individual model outputs and combines these probabilities using an aggregation. This technique leverages the uncertainty inherent in predictions and can result in improved overall performance and calibrated probability estimates.

We believe that our approach belongs to the third category where we aim to derive the relevant dependency among models (meta-model) using the graph-based representation. In the following, we will go in depth for this category. Our perspective aligns with the third class, wherein our objective is the extraction of pertinent inter-dependencies inherent in models (termed as the meta-model) through the utilization of a representation based on graphs. Subsequently, we intend to delve comprehensively into the intricacies of this particularly class.

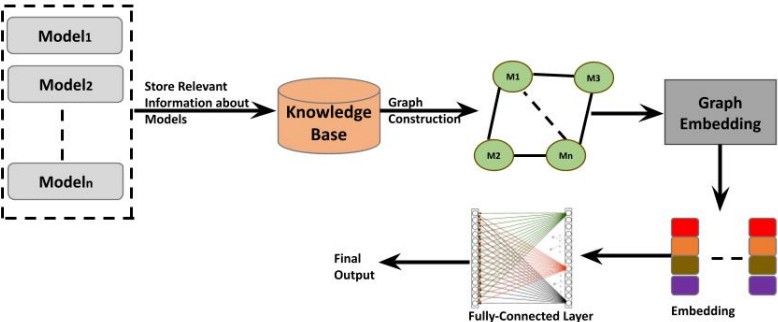

Figure 1: *LGR-ME's process starts by training machine learning models, we subsequently calculate the model specification, and output of each model. A model graph representation is created and learned using graph convolution neural network. A learned features are finally used to derive the final output of the desired task using fully connected layer.*

## 3 LGR-ME: LEARNING GRAPH REPRESENTATION FOR MODEL ENSEMBLE

### 3.1 PRINCIPLE

In this section, we introduce a novel approach termed "Learning Graph Representation for Model Ensemble" (LGR-ME), which combines principles from machine learning with graph-based representation to enhance model ensemble techniques. The conceptual framework of LGR-ME is illustrated in Figure 1. This approach involves several distinct stages, each contributing to the overall process of creating an integrated ensemble model. Initially, a diverse set of machine learning models is trained using appropriate datasets and algorithms. These models encompass various architectures and learning strategies. Upon completing the training phase, the resultant outputs and specifications of each individual model are stored within a knowledge base. This repository of model-specific information serves as the foundation for constructing a graph representation. The process of creating the model graph commences with the utilization of the accumulated knowledge base. Each trained

model is represented as a node in the graph, and the relationships between these nodes are established based on the observed interconnections and similarities among the models. This forms a graph where nodes denote the trained models, and edges signify the associations between these models, reflecting their shared characteristics and behaviors. Subsequently, a GCNN is deployed to glean insights from the constructed model graph. The primary objective of this step is to capture the intricate correlations and dependencies that exist between the various models. The GCNN operates to transform the information embedded within the model graph into a lower-dimensional vector space, effectively capturing the nuanced relationships that might not be apparent in the original high-dimensional space. The GCNN phase produces an embedded vector capturing the relationships among the trained models. This vector, a condensed representation of the ensemble's collective behaviors, is then processed by a fully connected layer to generate an output tailored to the user's objective. The subsequent sections detail LGR-ME's core components

## 3.2 Graph Representation for Models

**Definition 1** (Model Output). *We define the set of outputs of the model $\mathcal{M}_i$ by the union of all of its outputs when training the set of data in $D$, and we write:*

$$\mathcal{Y}_i^* = \{ \bigcup_{D_j \in D} y_{ij}^* \} \tag{1}$$

*where, $y_{ij}^*$ is the predicted value of $D_j$ by the model $\mathcal{M}_i$.*

**Definition 2** (Model Specification). *We define the set of model specification $\mathcal{S}_i$ of the model $\mathcal{M}_i$ by the set of the representative layers of the model $\mathcal{M}_i$. For instance, $\mathcal{S}_i = \{conv = 1, pool = 1, att = 0, bn = 0, dr = 1\}$ to represent that the model $\mathcal{M}_i$ has a convolution layer, a pooling layers, no attention layer, no batch normalization layer, and has a dropout layer.*

**Definition 3** (Connectivity Function). *Consider a set of $n$ models $\mathcal{M} = \{\mathcal{M}_1, \mathcal{M}_2...\mathcal{M}_n\}$, we define $f_{ij}$ a function that connect the two models $\mathcal{M}_i$, and $\mathcal{M}_j$, and we write:*

$$f : \mathcal{M} \times \mathcal{M} \rightarrow \mathbb{R}$$

**Definition 4** (Characteristic Connectivity Function ). *We define the Characteristic Connectivity Function (CCF) as a connectivity function that compute the similarity between two models specification, and we write:*

$$CCF(\mathcal{M}_i, \mathcal{M}_j) = |S_i \cap S_j| \tag{2}$$

**Definition 5** (Performance Connectivity Function). *We define the Performance Connectivity Function (PCF) as a connectivity function that compute the similarity between two models output, and we write:*

$$PCF(\mathcal{M}_i, \mathcal{M}_j) = |Y_i^* \cap Y_j^*| \tag{3}$$

**Definition 6** (Graph of Models). *We define the graph of models $\mathcal{G}_M = (V, E)$ by the set of nodes $V$ represented the set of models $\mathcal{M}$, and a set of edges, $E \subseteq V \times V$. $(\mathcal{M}_i, \mathcal{M}_j) \in E$, if and only if, $f_{ij} \neq 0$. $f_{ij}$ should be even $CCF(\mathcal{M}_i, \mathcal{M}_j)$ or $PCF(\mathcal{M}_i, \mathcal{M}_j)$ depending to the connectivty function chosen by the user.*

## 3.3 Learning Algorithm

Algorithm 1 presents the LGR-ME, it is designed to exploit the relationships among various models, which is crucial for ensemble methods and feature selection in machine learning. The algorithm takes a set of models $\mathcal{M}$ and training data points $\mathcal{D}$ as input and outputs a trained ensemble model. The LGR-ME algorithm begins by initializing an empty graph $G$ to represent the relationships between different models in $\mathcal{M}$ (Line 1). For each unique pair of models, it computes a weight using a function $f$ and adds a corresponding edge to the graph $G$ (Lines 3-6). Subsequently, a Maximum Spanning Tree (MST) is extracted from $G$ to represent the backbone of the most significant relationships between the models (Line 8). The algorithm then computes the Laplacian matrix $\mathcal{L}$ from the MST, which is used to derive the Laplacian loss $L_{\text{laplace}}$ (Lines 10-12). The GCNN model is initialized and trained using a newly introduced hybrid loss function (Lines 14-18). An important contribution of the LGR-ME algorithm is the introduction of this new loss function, $L(x)$, defined as:

$$L(x) = C(x) + \lambda D(x) + \gamma L_{\text{laplace}} \tag{4}$$

where $C(x)$ is the Cross-Entropy loss, $D(x)$ is the Kullback-Leibler Divergence loss, $L_{\text{laplace}}$ is the Laplacian loss ensuring the model's predictions align with the underlying graph structure, and $\lambda$ and $\gamma$ are regularization parameters.

This loss function combines the benefits of accuracy, diversity, and structural alignment, making it a versatile intuition.

---

**Algorithm 1** LGR-ME Algorithm

**Data:** Set of models $\mathcal{M}$, Training data points $\mathcal{D}$
**Result:** Trained LGR-ME model

1   $G \leftarrow \emptyset$                                         `// Initialize the graph`
2   **foreach** $(m_i, m_j) \in \mathcal{M}^2 \ s.t \ i < j$ **do**
3       $w_{ij} = f(m_i, m_j)$          `// Compute weight between models using` $f$
4       $G \leftarrow G \cup \{(m_i, m_j, w_{ij})\}$         `// Add edge with weight to the graph`
5   $MST \leftarrow$ Extract Maximum Spanning Tree from $G$
6   $\mathcal{L} \leftarrow (d - A_{MST})$     `//` $d$ `is the diagonal degree matrix and` $A_{MST}$ `is the adjacency matrix`
7   $L_{\text{laplace}} \leftarrow \frac{1}{\sum_{i,j} \mathcal{L}_{ij}^2}$
8   Initialize GCNN model with parameters $\theta$
9   $L(x) = C(x) + \lambda D(x) + \gamma L_{\text{laplace}}$
10   $\theta_{t+1} = \theta_t - \eta \nabla L(\theta_t)$
11   **return** Trained GCNN model with parameters $\theta$

---

### 3.4 MAXIMUM SPANNING TREE FOR ENSEMBLE LEARNING

1. **Relationships Capture**: **Statement** The maximum spanning tree tends to capture the most significant relationships among nodes. In our context, using edge weights that describe shared properties means that the maximum spanning tree would emphasize the most relevant shared properties among the models.

   *Logical Reasoning:* The maximum spanning tree algorithm aims to select edges with the highest weights to create a tree that spans all nodes without forming cycles. By using edge weights that represent shared properties, the algorithm would naturally prioritize edges that connect nodes with the most similar properties. In the context of models, these edges would represent the most significant relationships based on shared properties. The maximum spanning tree focuses on capturing the most relevant and informative relationships among the models, leading to a meaningful representation of their connections.

2. **Reduced Redundancy**: **Statement** The maximum spanning tree selects edges to form a tree structure without creating cycles. This can help in reducing redundancy and focusing on the most essential relationships.

   *Logical Reasoning:* The structure of a tree inherently ensures that no cycles are formed. When the maximum spanning tree algorithm is applied to a graph, it aims to create a tree that connects all nodes with the highest-weighted edges without forming cycles. By avoiding cycles, the algorithm prevents redundant relationships from being included in the tree. This results in a focused structure that captures the most essential and informative relationships while avoiding unnecessary redundancy.

3. **Hierarchical Information Statement** If the maximum spanning tree has a hierarchical structure, it could provide insights into a hierarchy of shared properties among the models.

   *Logical Reasoning:* The hierarchical structure of a maximum spanning tree is influenced by the arrangement of edge weights and the properties they represent. If the shared properties have a hierarchical nature, the maximum spanning tree might naturally reflect this hierarchy. In cases where certain shared properties are more general and others are more specific, the algorithm would tend to connect nodes with similar general properties first, followed by

those with more specific properties. This hierarchy in the spanning tree can provide insights into the relationships among shared properties and their relevance to the models.

4. **Ensemble Composition**: **Statement** By considering only the most important relationships based on the maximum spanning tree, we might achieve a more focused and diverse ensemble of models.

   *Logical Reasoning:* The maximum spanning tree focuses on capturing the most significant relationships among models based on shared properties. By considering only the most important and informative relationships, the resulting ensemble can be more focused on leveraging the strengths of these relationships. The ensemble would include models that are connected through the most relevant shared properties, ensuring that the selected models complement each other's strengths. This can lead to a diverse ensemble that benefits from the most important aspects of each model's contribution.

## 3.5 Intuition of the loss function in Ensemble Learning

**Diversity and Accuracy**
The Cross-Entropy loss ensures that the ensemble is accurate, while the Kullback-Leibler Divergence can enforce diversity among the models.

**Regularization**
The $\lambda$ term acts as a regularization parameter. A higher $\lambda$ gives more importance to the Kullback-Leibler Divergence, encouraging the ensemble to pay more attention to distributional aspects.

**Laplacian Loss**
The Laplacian loss, derived from the Laplacian matrix of the graph, encourages the model to respect the graph's structure. In the context of spanning trees, this loss ensures that the model's predictions align with the underlying tree structure, promoting a more structured prediction.

**Trade-off**
The hybrid loss allows for a balance between classification accuracy, distributional similarity, and graph structure adherence, offering the 'best of both worlds.'

**Robustness**
The hybrid loss can make the ensemble more robust. While Cross-Entropy loss ensures accuracy, the Kullback-Leibler part can make the ensemble robust to variations in the input distribution.

**Optimal Weighting**
The hybrid loss can provide a nuanced performance metric, allowing for potentially more effective weighting of individual models in the ensemble.

## 3.6 Proof of Convexity and Convergence of the loss function

**Theorem 1.** *The hybrid loss function $L(x) = C(x) + \lambda D(x) + \gamma L_{laplace}$ is convex if $\lambda \geq 1$ and $\gamma$ is non-negative.*

*Proof.* Let $C(x)$ be the cross-entropy loss and $D(x)$ the Kullback-Leibler Divergence loss. To prove convexity, we examine the second derivatives, forming the Hessian matrix for $L(x)$. Let $p_i$ represent the true probability distribution for class $i$ and $q_i$ represent the predicted probability distribution for class $i$.

**Cross-Entropy Loss** $C(x)$:
$$C(x) = -\sum_i p_i \log(q_i)$$

**Kullback-Leibler Divergence** $D(x)$:
$$D(x) = \sum_i p_i \log\left(\frac{p_i}{q_i}\right)$$

To build the Hessian matrix we need to find the mixed partial derivative $\frac{\partial^2 C(x)}{\partial q_i \partial q_j}$. Since each term $p_i \log(q_i)$ (and $p_i \log(\frac{p_i}{q_i})$) only involves $q_i$ and no other $q_j$ where $i \neq j$, the mixed partial derivative

is zero:

$$\frac{\partial^2 C(x)}{\partial q_i \partial q_j} = 0$$

For both $C(x)$ and $D(x)$, the Hessian matrices are diagonal. Therefore, we only need to consider the second derivative with respect to each $q_i$ or $p_i$.

- Second derivative of $C(x)$ with respect to $q_i$: $-\frac{p_i}{q_i^2}$.

- Second derivative of $D(x)$ with respect to $q_i$: $-\frac{p_i}{q_i^2}$.

**Laplacian Loss $L_{\text{laplace}}$:**

$$L_{\text{laplace}} = x^T \mathcal{L} x$$

The Hessian of $L_{\text{laplace}}$ is simply the Laplacian matrix $\mathcal{L}$ itself. The Laplacian matrix is positive semi-definite, which means all its eigenvalues are non-negative. This ensures that the Laplacian loss is convex.

**Total Loss $L(x)$:**

$$L(x) = C(x) + \lambda D(x) + \gamma L_{\text{laplace}}$$

Second derivative with respect to $q_i$: $-\frac{p_i(1+\lambda)}{q_i^2}$.

The function $L(x)$ will be convex if $-\frac{p_i(1-\lambda)}{q_i^2} \geq 0$ for all $i$, and the Laplacian loss is convex.

- $p_i \geq 0$ because $p_i$ is a probability.

- $q_i > 0$ because $q_i$ is a probability.

- $1 - \lambda \leq 0$ implies $\lambda \geq 1$.

Therefore, if $\lambda \geq 1$ and $\gamma$ is non-negative, then $L(x)$ is convex, as all the second derivatives are non-negative, making the Hessian matrix positive semi-definite.

Thus, the loss function $L(x) = C(x) + \lambda D(x) + \gamma L_{\text{laplace}}$ is convex when $\lambda \geq 1$ and $\gamma$ is non-negative. $\qquad \square$

**Theorem 2.** *If the gradients of the loss function $L(x) = C(x) + \lambda D(x) + \gamma L_{laplace}$ are bounded, then gradient-based optimization methods like gradient descent will converge.*

*Proof.* To prove that the gradients are bounded, we examine the first derivatives of $L(x)$.

**Cross-Entropy Loss $C(x)$:**

$$C(x) = -\sum_i p_i \log(q_i)$$

First derivative with respect to $q_i$: $-\frac{p_i}{q_i}$.

**Kullback-Leibler Divergence $D(x)$:**

$$D(x) = \sum_i p_i \log\left(\frac{p_i}{q_i}\right)$$

First derivative with respect to $q_i$: $-\frac{p_i}{q_i}$.

**Total Loss $L'(x)$:**

$$L'(x) = C(x) + \lambda D(x)$$

First derivative with respect to $q_i$: $-\frac{p_i(1+\lambda)}{q_i}$.

Since $p_i$ and $q_i$ are probabilities, they are bounded between 0 and 1. Also, $\lambda$ is a non-negative constant. Therefore, the first derivative of $L'(x)$ is bounded.

The gradient of the Laplacian loss $L_{\text{laplace}}$ with respect to $x$ is:

$$\nabla L_{\text{laplace}} = 2\mathcal{L}x$$

Given that $\mathcal{L}$ is the Laplacian matrix, which is derived from the adjacency matrix of the graph, its values are bounded. This ensures that the gradient of $L_{\text{laplace}}$ is bounded.

Now, combining the boundedness of the gradients of Cross-Entropy loss, Kullback-Leibler Divergence, and the Laplacian loss, we can conclude that the gradients of the loss function $L(x) = C(x) + \lambda D(x) + \gamma L_{\text{laplace}}$ are bounded. By the boundedness of the gradients, we can conclude that gradient-based optimization methods like gradient descent will converge when optimizing $L(x)$. □

## 4 EXPERIMENTATION

### 4.1 EXPERIMENTAL SETUP

We evaluated the LGR-ME algorithm on five Kaggle classification datasets: Employee, Heart-attack, Titanic, Credit Card Approval, and Water Potability Inc. (2023). Datasets underwent preprocessing, normalization, and encoding. The graph's representation was built using a diverse set of classifiers with tailored hyperparameters. These include **Random Forest** with 10 estimators and a minimum samples split of 2, **Gradient Boosting** with 10 estimators and a 0.1 learning rate, **Logistic Regression** using L2 penalty and 'lbfgs' solver, **SVC** with an RBF kernel and 'scale' gamma, **K-Nearest Neighbors** with 5 neighbors, **Decision Tree** using Gini criterion, **Gaussian Naive Bayes** with a variance smoothing of $1 \times 10^{-9}$, and **MLP** with a 100-neuron hidden layer. Only the $PCF$ function is used to assign weights to the graph edges. Our GCNN processes graph data with three graph convolutional layers, transforming node features into 128, 32, and 18-dimensional spaces. Post-aggregation, features are averaged and passed through a fully connected layer, producing class probabilities.The GCNN is trained for 50 epochs with a batch size of 128 with a regularization parameters $\lambda = \gamma = 5.0$. To ensure a robust evaluation of the GCNN model, we employed KFold cross-validation where $K = 5$ .

### 4.2 RESULTS DISCUSSION

The LGR-ME algorithm's performance across various datasets underscores its ability to adeptly combine the strengths of multiple weak classifiers. LGR-ME often results in achieving superior performance metrics compared to any single classifier in its ensemble.

For the Water Potability dataset for instance, LGR-ME's performance is nothing short of remarkable, achieving an accuracy of 0.9975, which is significantly higher than any other model, including RandomForest with an accuracy of 0.6368 and GradientBoosting at 0.6275. This demonstrates LGR-ME's capability to handle intricate patterns and nuances in datasets.

The convergence of the loss function is a crucial indicator of the stability and reliability of a learning algorithm. For the datasets under consideration, we observed consistent convergence patterns, as illustrated in figure 2 below. For each of the highlighted datasets, the loss function exhibits a clear trend towards convergence, indicating that the model is learning the underlying patterns effectively. The detailed results and further discussions are provided in the appendix.

| Model | Accuracy | Precision | Recall | F1 Score |
|---|---|---|---|---|
| RandomForestClassifier | 0.6368 | 0.6261 | 0.6368 | 0.6114 |
| GradientBoostingClassifier | 0.6275 | 0.6149 | 0.6275 | 0.6033 |
| LogisticRegression | 0.5958 | 0.4742 | 0.5958 | 0.4481 |
| SVC | 0.4291 | 0.5044 | 0.4291 | 0.3653 |
| KNeighborsClassifier | 0.5516 | 0.5316 | 0.5516 | 0.5354 |
| DecisionTreeClassifier | 0.5964 | 0.5967 | 0.5964 | 0.5963 |
| GaussianNB | 0.6169 | 0.5993 | 0.6169 | 0.5752 |
| MLPClassifier | 0.5629 | 0.6245 | 0.5629 | 0.4230 |
| LGR-ME | **0.9975** | **0.9979** | **0.9979** | **0.9979** |

Table 1: Average Metrics Over All Folds for Water Potability dataset

On the Titanic dataset, LGR-ME again showcases its prowess with an accuracy of 0.9655, outperforming other models such as RandomForest (0.7331) and GradientBoosting (0.7469). Similarly, for the Employee dataset, LGR-ME achieves an accuracy of 0.8936, surpassing RandomForest (0.8197) and DecisionTree (0.8114).

| Model | Accuracy | Precision | Recall | F1 Score |
|---|---|---|---|---|
| RandomForestClassifier | 0.7331 | 0.7328 | 0.7331 | 0.7285 |
| GradientBoostingClassifier | 0.7469 | 0.7435 | 0.7469 | 0.7390 |
| LogisticRegression | 0.7120 | 0.6961 | 0.7120 | 0.6973 |
| SVC | 0.6094 | 0.3891 | 0.6094 | 0.4734 |
| KNeighborsClassifier | 0.6775 | 0.6778 | 0.6775 | 0.6743 |
| DecisionTreeClassifier | 0.7605 | 0.7726 | 0.7605 | 0.7605 |
| GaussianNB | 0.6297 | 0.6725 | 0.6297 | 0.6218 |
| MLPClassifier | 0.6299 | 0.6030 | 0.6299 | 0.6097 |
| **LGR-ME** | **0.9655** | **0.9717** | **0.9586** | **0.9650** |

Table 2: Average Metrics Over All Folds for Titanic dataset

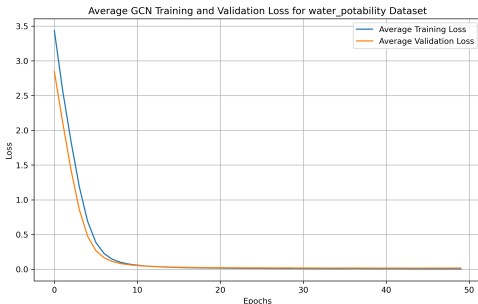

Figure 2: Training/Validation Loss of Water potability averaged for all the folds

## 5 CONCLUSION

We introduce in this paper a novel approach, Learning Graph Representation for Model Ensemble (LGR-ME), which integrates foundational machine learning principles with the construct of a graph representation. It involves training and storing the outputs of diverse machine learning models in a knowledge base, creating a graph that signifies model relationships, applying a GCNN to learn complex dependencies, and ultimately utilizing a fully connected layer to yield the final learning output aligned with the user's desired task. This representation is then subjected to training by a meta-model, which seeks to minimize a novel loss function we have defined. This combined loss effectively took into account both diversity, accuracy and the concept of maximum spanning trees. Additionally, we provided a comprehensive theoretical examination of the newly introduced loss function. This intricate and comprehensive methodology offers a powerful means of harnessing the collective capabilities of diverse models to enhance ensemble learning outcomes. Its efficacy is showcased in classification task on different benchmarks. The potential of LGR-ME has been demonstrated in specific classification learning task. Our aim is to amplify its applicability across a wider range of machine learning scenarios. This endeavor entails customizing the framework to accommodate various data domains, types of problems, and assessment criteria. To illustrate, LGR-ME has the potential to be employed in contexts like reinforcement learning, tasks within natural language processing, and even intricate challenges in bioinformatics.

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

# A APPENDIX

## A.1 REMAINING RESULTS

It's essential to note that LGR-ME's performance isn't universally dominant. For instance, on the Heart-attack dataset, GaussianNB achieves the highest accuracy of $0.8142$, while LGR-ME lags behind with $0.6401$. This suggests that while LGR-ME is generally adept at combining weak classifiers, its performance can still be contingent on the nature of the dataset and the quality of the base classifiers.

For the Credit Card Approvals dataset, LGR-ME isn't the top performer, with GCN taking the lead at $0.9074$. However, LGR-ME's ensemble approach still allows it to achieve competitive results across the board, emphasizing the power of collaborative learning.

| Model | Accuracy | Precision | Recall | F1 Score |
|---|---|---|---|---|
| RandomForestClassifier | 0.8197 | 0.8174 | 0.8197 | 0.8148 |
| GradientBoostingClassifier | 0.8361 | 0.8373 | 0.8361 | 0.8296 |
| LogisticRegression | 0.7117 | 0.7015 | 0.7117 | 0.6808 |
| SVC | 0.5035 | 0.6116 | 0.5035 | 0.5016 |
| KNeighborsClassifier | 0.7800 | 0.7764 | 0.7800 | 0.7693 |
| DecisionTreeClassifier | 0.8114 | 0.8085 | 0.8114 | 0.8082 |
| GaussianNB | 0.6924 | 0.6809 | 0.6924 | 0.6831 |
| MLPClassifier | 0.6150 | 0.5261 | 0.6150 | 0.5259 |
| LGR-ME | **0.8936** | **0.8932** | **0.8932** | **0.8932** |

Table 3: Average Metrics Over All Folds for Employee dataset

| Model | Accuracy | Precision | Recall | F1 Score |
|---|---|---|---|---|
| RandomForestClassifier | 0.7727 | 0.7779 | 0.7727 | 0.7730 |
| GradientBoostingClassifier | 0.7440 | 0.7517 | 0.7440 | 0.7439 |
| LogisticRegression | 0.8140 | 0.8166 | 0.8140 | 0.8133 |
| SVC | 0.6662 | 0.6782 | 0.6662 | 0.6562 |
| KNeighborsClassifier | 0.6781 | 0.6877 | 0.6781 | 0.6752 |
| DecisionTreeClassifier | 0.7484 | 0.7562 | 0.7484 | 0.7482 |
| GaussianNB | **0.8142** | **0.8188** | **0.8142** | **0.8138** |
| MLPClassifier | 0.6859 | 0.6698 | 0.6859 | 0.6661 |
| LGR-ME | 0.6401 | 0.6401 | 0.6401 | 0.6401 |

Table 4: Average Metrics Over All Folds for Heart-attack dataset

| Model | Accuracy | Precision | Recall | F1 Score |
|---|---|---|---|---|
| RandomForestClassifier | 0.8747 | 0.8793 | 0.8747 | 0.8748 |
| GradientBoostingClassifier | 0.8166 | 0.8180 | 0.8166 | 0.8164 |
| LogisticRegression | 0.8348 | 0.8389 | 0.8348 | 0.8334 |
| SVC | 0.6115 | 0.7108 | 0.6115 | 0.5269 |
| KNeighborsClassifier | 0.6787 | 0.6815 | 0.6787 | 0.6756 |
| DecisionTreeClassifier | 0.8220 | 0.8242 | 0.8220 | 0.8218 |
| GaussianNB | 0.7985 | 0.8140 | 0.7985 | 0.7926 |
| MLPClassifier | 0.7839 | 0.7913 | 0.7839 | 0.7825 |
| **GCN** | **0.9074** | **0.9074** | **0.9074** | **0.9074** |

Table 5: Average Metrics Over All Folds for Credit Card Approvals dataset

For all the datasets, the loss function exhibits a clear trend towards convergence, indicating that the model is learning the underlying patterns effectively.

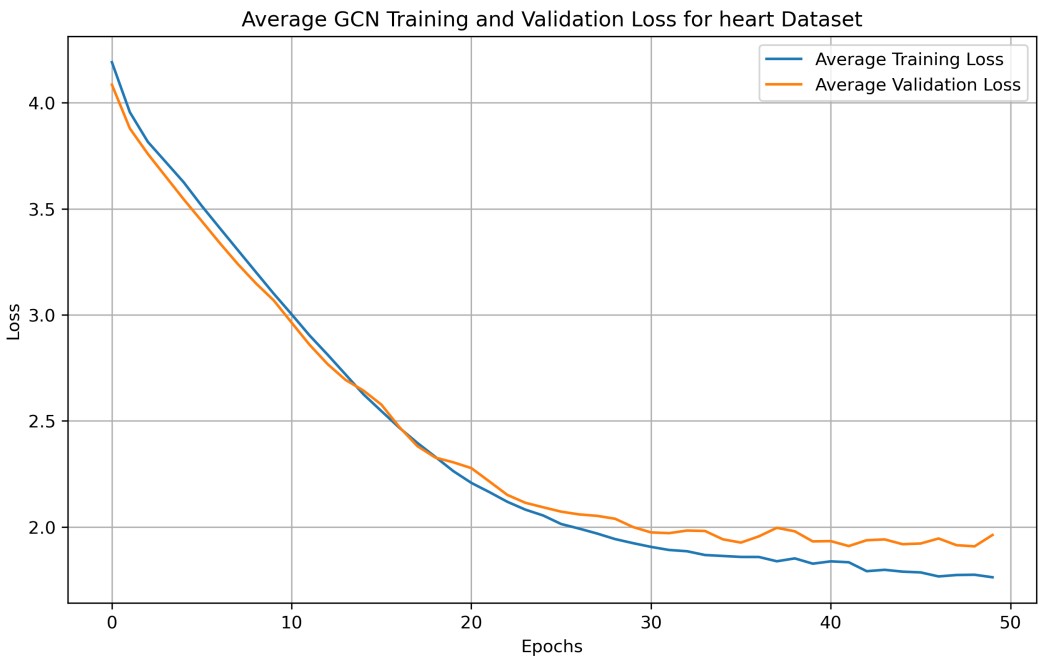

Figure 3: Training/Validation Loss for Heart Attack dataset

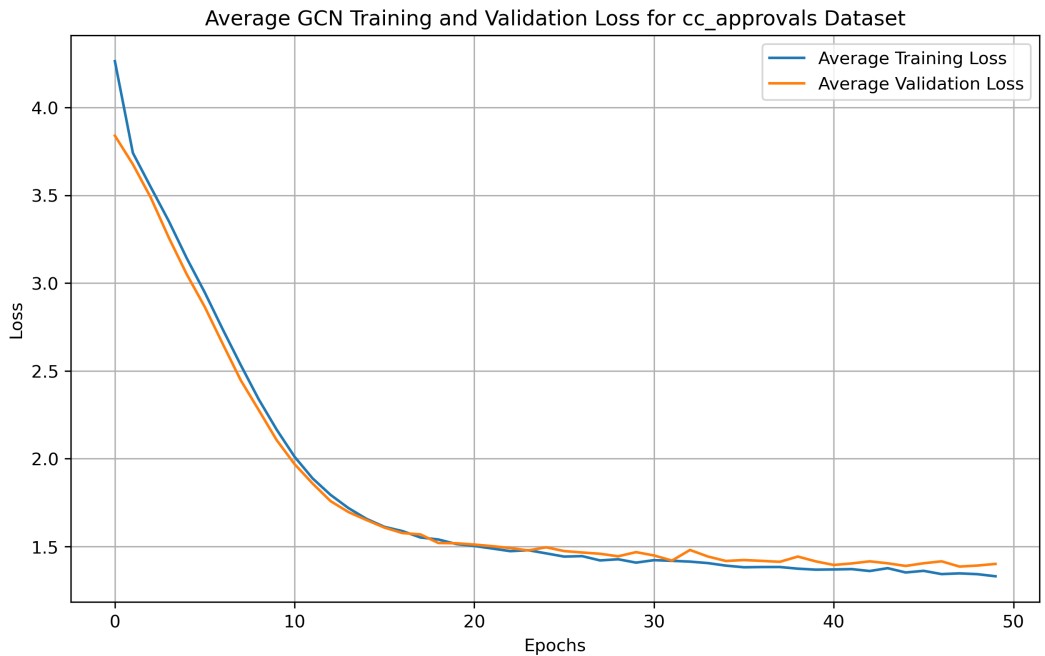

Figure 4: Training/Validation Loss for Credit Card Approvals dataset

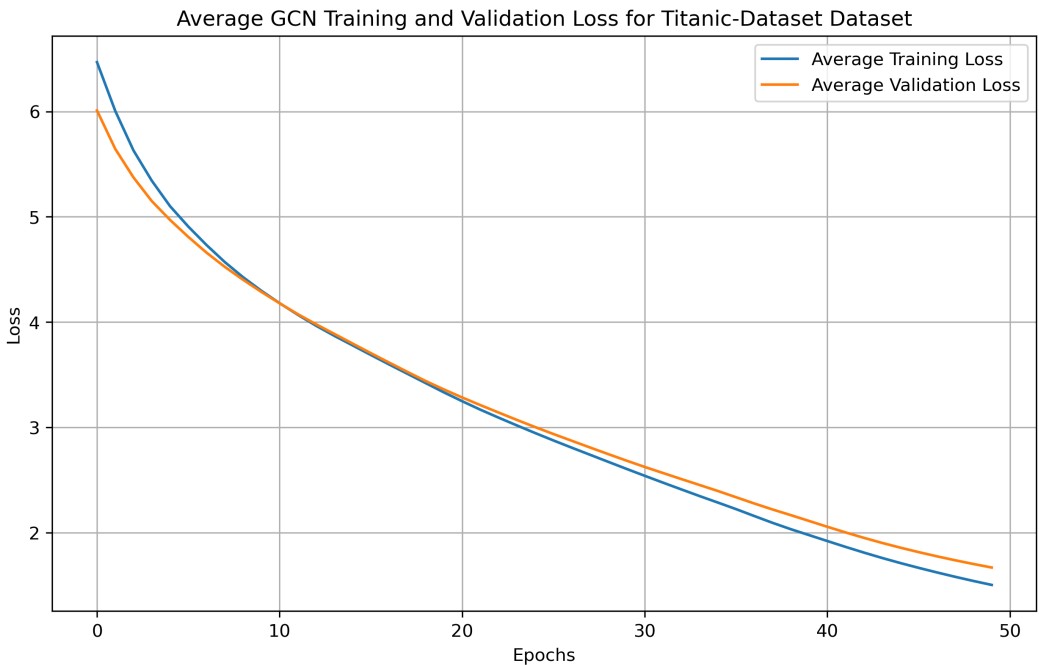

Figure 5: Training/Validation Loss for Titanic dataset

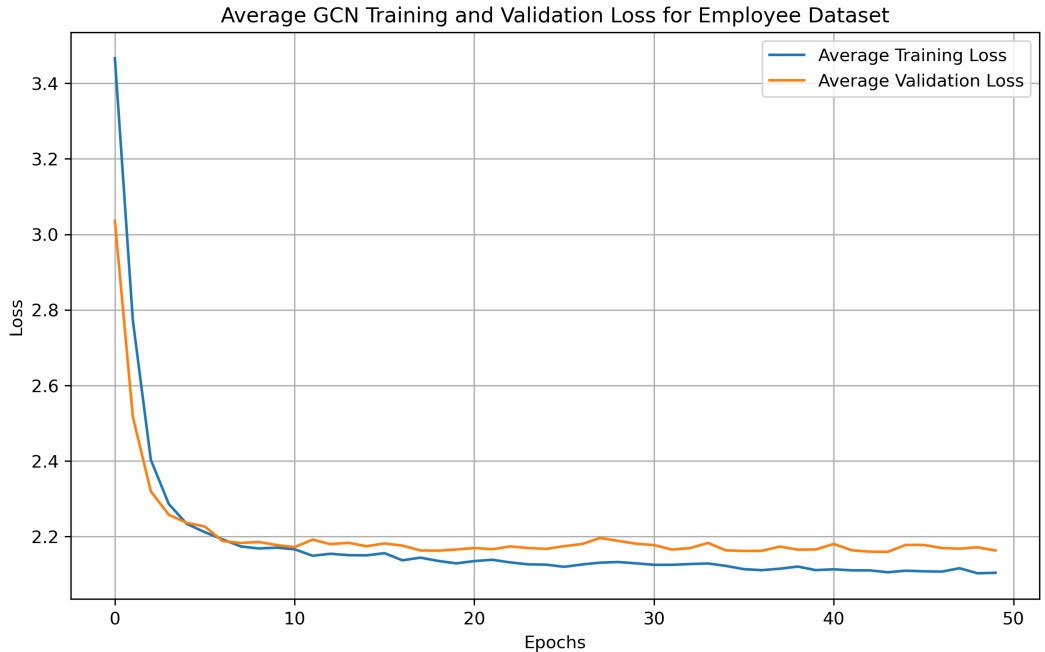

Figure 6: Training/Validation Loss for Emplyee dataset

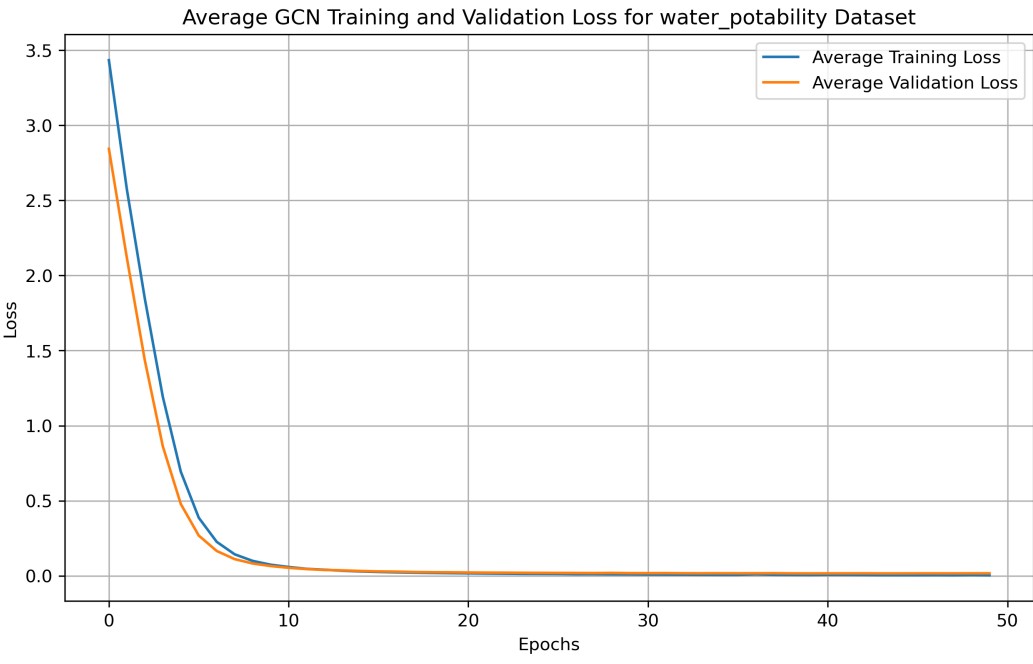

Figure 7: Training/Validation Loss for Water Potability dataset

