# OpenReview forum: "Learning Graph Representation for Model Ensemble"
_ICLR.cc/2024/Conference — ICLR 2024 Conference Withdrawn Submission_

### Official Review · Reviewer_3xiQ · 2023-10-17

**Soundness:** 1 poor
**Presentation:** 1 poor
**Contribution:** 1 poor
**Rating:** 1
**Confidence:** 5

**Summary:**

The paper explores an ensemble strategy for neural networks, which exploits a graph NN to perform the final prediction. The key steps of the algorithm are: (1) they build a graph where the nodes are the models and the edges depend either on the overlap across predictions of the networks, or on their architectural similarity; (2) they apply a GNN on top of this graph; (3) they optimize a loss composed of three terms (a cross-entropy, a KL divergence for "diversity", and a Laplacian smoothness term). They show some results on some simple tabular datasets.

**Strengths:**

The only "strength" I can think of is that the topic is interesting, and I like the idea of introducing a graph relation across models to improve the ensemble. However, the way the idea is executed in the paper is not acceptable for any scientific conference. Even addressing all questions below, I still feel the scientific novelty would not warrant presentation at ICML.

**Weaknesses:**

There are many weaknesses and questions so I will focus on the key ones.

1. Claims made in the paper are excessive and the paper should be significantly rewritten, e.g., "groundbreaking approach", "general-purpose learning systems", "foundational framework", "self-adaptation", "a foundational paradigm for creating adaptable machine learning systems", etc. The method is an ensemble technique for neural networks, and it does not deviate too much from other ensemble / merging / blending strategies. Abstract and introduction should reflect this.

2. Always from the introduction, "The foremost challenge emanates from their resource-intensive nature", but the model does not address this since running the graph requires running the entire ensemble. "by selecting the optimal subset" is another sentence that is not justified since the graph is constructed deterministically and not trained. "overcome limitations found in both dedicated single-purpose models and multipurpose models tailored for specific tasks" is yet another example of how claims are excessive, since the model must be retrained for each task.

3. Many key parts of the method are unclear or completely left out of the paper. These include:

(a) "Characteristic Connectivity Function" should measure the similarity between models but it seems too restrictive (e.g. a model with 2 conv layers and 1 dense layer against a model with 1 conv layer and 2 dense layers would have a CCF of 0?).

(b) The authors are never stating what is the input of the GCN (predictions? embeddings from the last layer?).

(c) Are the original models fine-tuned in the last stage? Are they frozen?

(d) "Kullback-Leibler Divergence" in (4) should guarantee the "diversity" of the models, but it is the KL divergence computed on what? The predictions of the models?

(e) All the components are never benchmarked in isolation (e.g., do you need the spanning tree? Does the Laplacian smoothness term improves the results? Etc.)

(f) Algorithms for the comparison are very standard ML models. There is a huge literature on ensembling / merging neural networks which is only briefly touched in the paper and never benchmarked.

(g) The results are questionable. They only use toy datasets, and in some cases the results are very strange (e.g., Titanic score can easily get > 80% with any model, see https://www.kaggle.com/code/alexisbcook/titanic-tutorial). Their models have sometimes > 20% absolute improvement against strong baselines such as random forest.

(h) Theorems 1 and 2: these prove convexity of the loss (with respect to the predictions). But this is just a combination of cross-entropy and KL, and the overall model would be non-convex anyway since we are using neural networks. The entire analysis here is lacking value.

**Questions:**

The questions follow more or less the drawbacks:

3a: provide more details on how the connectivity is built.

3b: provide the precise equations or structure of the model.

3c: clarify the training process in pseudo-code.

3d-e: provide clear ablation studies on all components of the model.

3f-g: add more baselines and datasets to the experimental comparison, and carefully check the results and hyper-parameters of all models.

3h: remove the "theoretical analysis" if unneeded.

---

### Official Review · Reviewer_oYx9 · 2023-10-31

**Soundness:** 2 fair
**Presentation:** 1 poor
**Contribution:** 1 poor
**Rating:** 1
**Confidence:** 3

**Summary:**

This paper proposes a learning strategy called LGR-ME for model ensemble. The authors define a characteristic connectivity function and a performance connectivity function to calculate the degree of connection between models. Then, a graph is constructed by the calculation and maximum spanning tree (MST) is extracted from the graph. MST is used to derive the Laplacian loss based on the connectivity between models. The total loss is defined as the sum of the cross-entropy loss, KL divergence loss, and Laplacian loss.

**Strengths:**

It is hard to find strentghs of this paper.

**Weaknesses:**

1. The citation format is incorrect. When the authors or the publication are not included in the sentence, "\citep" should be used.

2. The detailed explanation is severely lacking. For example, how is the similarity between two model specifications calculated? How is the similarity between two model outputs calculated?

3. The proposed method is only compared with outdated techniques (e.g., Random Forest, SVC, and MLP). Moreover, it should be compared with ensemble strategies, not classifiers.

4. The authors state, "We introduce, LGR-ME (Learning Graph Representation for Model Ensemble), a groundbreaking approach within the domain of general-purpose learning systems." However, the authors only evaluate their methods for classification on simple datasets.

**Questions:**

Q1. How is the similarity between two model specifications calculated?

Q2. How is the similarity between two model outputs calculated?

Q3. Could the authors compare their method with other ensemble learning methods?

---

### Official Review · Reviewer_WLfx · 2023-11-01

**Soundness:** 2 fair
**Presentation:** 1 poor
**Contribution:** 2 fair
**Rating:** 3
**Confidence:** 4

**Summary:**

This paper studies the model ensemble using Graph Neural Networks (GNNs). Specifically, they regard each model (specification) in the pool as node. Also, they define two connectivity functions to construct the edges. And then, they employ a graph neural network (GCNN) to  learn complex dependencies among various models, and ultimately utilizing a fully connected layer to yield the final learning output aligned with the user’s desired task. Experiments verify the effectiveness of the proposed method.

**Strengths:**

[+] The idea of formulating model ensemble is interesting.
[+] Extensive experiments are performed to verify the effectiveness of the proposed method.
[+] The codes are provided for reproducing the results.

**Weaknesses:**

[-] Are the model specifications adequate to describe the models? For instance, the authors claimed that they use Si = {conv = 1, pool = 1, att = 0, bn = 0, dr = 1} to represent that the model Mi has a convolution layer, a pooling layers, no attention layer, no batch normalization layer, and has a dropout layer. However, a plenty of important information (the dimension of each layer, the dropout ratio, and etc.) of the model is ignored.

[-] In the experiments, how does the  model specifications of other models (Random Forest, Gradient Boosting, and etc.) look like? It is extremely important to provide these experimental details.

[-] The authors only compare with single model. But, can the proposed method perform better than advanced ensemble methods [1]? Especially, some recent works on model fusion [2,3] exhit overwhelming superiority over previous methods.

[-] I notice that the authors only adopt the Performance Connectivity Function (PCF) to construct the edges. So, is it necssary to define the Characteristic Connectivity Function (CCF) in the paper?  Actually, CCF would be an empty set when the two models (e.g., Random forest and MLP) are completely different.

[-] The authors adopt 3-layer GNNs in the experiments. However, there are only 7 nodes (models) in the experiments. The well-known oversmoothing issue of GNNs would degrade the performance given that the deep GNN layer and few nodes.

[-] The presentation id not clear. For example, I suggest that the authors to provide detailed formulations of $C(x)$ , $D(x)$, and $L_{laplace}$ after Eq. (4).

[-] Please use the symbols consistently. For example, in Figure 1, is Model1 the same as M1? The $\mathcal{Y}$ in Eq. (1) should be same as $Y$ in Eq. (3).

[-] The citation style of the paper is wrong. I expect the authors can use \citep{} and \citet{} correctly.

[-] The quotation marks (e.g., ’best of both worlds.’) in this manuscript are wrong.


[1] A Survey on Ensemble Learning under the Era of Deep Learning (arXiv:2101.08387)
[2] Model Fusion via Optimal Transport (NeurIPS 2020)
[3]  Git Re-Basin: Merging Models modulo Permutation Symmetries (ICLR 2023)

**Questions:**

1. Are the model specifications adequate to describe the models? For instance, the authors claimed that they use Si = {conv = 1, pool = 1, att = 0, bn = 0, dr = 1} to represent that the model Mi has a convolution layer, a pooling layers, no attention layer, no batch normalization layer, and has a dropout layer. However, a plenty of important information (the dimension of each layer, the dropout ratio, and etc.) of the model is ignored.

2. In the experiments, how does the  model specifications of other models (Random Forest, Gradient Boosting, and etc.) look like? It is extremely important to provide these experimental details.

3. he authors only compare with single model. But, can the proposed method perform better than advanced ensemble methods?

4. Is it necssary to define the Characteristic Connectivity Function (CCF) in the paper?

---

### Official Review · Reviewer_eLAm · 2023-11-01

**Soundness:** 1 poor
**Presentation:** 2 fair
**Contribution:** 1 poor
**Rating:** 1
**Confidence:** 3

**Summary:**

This paper introduces an ensembling method where a number of models are trained for a specific task and then a graph of these models is created based on similarity in model specification and output. Then a GCN is trained on a maximum spanning tree of this graph to produce the final model.

**Strengths:**

Ensembling can be important for improving the performance of machine learning models and some regularizations like dropout implicitly perform ensembling.

**Weaknesses:**

There are several things that are vague and confusing about this submission.  First the loss function does not make sense to me. If the loss is  L(x) = C(x) +\lambda D(x) + \gamma L_laplace
Then the D(x) KL divergence would be minimized rather than maximized which is what the authors state they want for diversity. The Laplacian of the graph would not change unless the underlying graph changes so it is unclear what adding this to the loss function does.

The writing of this manuscript is also very unusual with the formating of Section 3.4 being very unclear.

The results are also confusing because most of them are not trying to ensemble generic ML models. For example random forest only contains a set of decision trees, and KNN classifiers are not an ensemble method. I don't see any comparisons to an ensemble of deep learning or other machine learning models.

**Questions:**

Can you explain the loss functions and experimental setup better?